# Betatrophin and Insulin Resistance

**DOI:** 10.3390/metabo12100925

**Published:** 2022-09-29

**Authors:** Qi Guo, Shicheng Cao, Xiaohong Wang

**Affiliations:** 1Department of Health Promotion, School of Intelligent Medicine, China Medical University (CMU), Shenyang 110000, China; 2Center of 3D Printing & Organ Manufacturing, School of Intelligent Medicine, China Medical University (CMU), Shenyang 110000, China

**Keywords:** betatrophin, T2D, exercise

## Abstract

Betatrophin (angiopoietin-like protein 8 (ANGPTL8)) is a hormone that was recently discovered in the human liver. Multiple homologous sequences have been detected in mammalian liver, white adipose, and brown adipose tissues. Betatrophin is crucial for the development of type 2 diabetes (T2D), insulin resistance, and lipid metabolism. Similar to the intake of insulin, thyroid hormones, irisin, and calories, betatrophin expression in the organism is usually attributed to energy consumption or heat generation. It can mediate the activity of lipoprotein lipase (LPL), which is the key enzyme of lipoprotein lipolysis. Due to its association with metabolic markers and the roles of glucose and lipid, the physiological function of betatrophin in glucose homeostasis and lipid metabolism can be more comprehensively understood. Betatrophin was also shown to facilitate pancreatic β-cell proliferation in a mouse model of insulin resistance. There are also reports that demonstrate that betatrophin regulates triglycerides (TGs) in the liver. Therefore, the process of regulating the physiological function by betatrophin is complicated, and its exact biological significance remains elusive. This study provides a comprehensive review of the current research, and it discusses the possible physiological functions of betatrophin, and specifically the mechanism of betatrophin in regulating blood glucose and blood lipids.

Betatrophin, also known as TD26, LPL inhibitor (lipasin), and ANGPTL8, is a protein that is primarily expressed in the liver and adipose tissues. Preliminary studies have confirmed that it mainly plays a role in lipid metabolism [1,2]. Recently, betatrophin has been shown to enhance glucose tolerance during insulin resistance, which is probably a result of the increased proliferation of pancreatic β-cells [3]. However, some subsequent experiments have questioned the conclusion that betatrophin enhances glucose tolerance and regulates the blood glucose balance by proliferating pancreatic β-cells, but the role of betatrophin in regulating lipid metabolism has been confirmed by some investigators [4,5,6,7]. Therefore, it appears that this protein is probably a promising target for the treatment of metabolic syndrome (MetS) and T2D. Diabetes prevalence is rising worldwide, and it currently affects around 415 million adults, and it is expected to affect around 640 million by 2040, according to estimates by the International Diabetes Federation (IDF). In addition, there is a large proportion of people with prediabetes, 50% of whom will develop diabetes within 10 years if there is no effective intervention with preventive measures [8]. Controlling the blood glucose by lifestyle intervention and the administration of hypoglycemic drugs and insulin can effectively prevent the occurrence and postpone the development of diabetes [9,10]. Using biomarkers to predict the occurrence, future development, and severity of diabetes, and its severe complications, is another measure that may significantly delay or prevent the occurrence of diabetes or its complications. In this review, betatrophin, a recently discovered biomarker, was the topic of interest. Despite rare reports on the physiological function and molecular target of betatrophin, various biological functions of betatrophin in the human body and mouse models have been recently confirmed, and this biomarker is considered to be an independent predictor of T2D development [11]. Therefore, the identification, cytological function, and regulatory mechanisms of betatrophin were investigated in this study.

## 1. Identification, Structure, Localization, and Secretion of Betatrophin

### 1.1. Identification of Betatrophin

Betatrophin, which was discovered in 2004, is a tumor-related specific serum antigen [12]. Since then, this new type of protein has received little attention. In 2012, betatrophin showed a positive correlation with serum TG and betatrophin overexpression by adenoviruses in mice that increased the serum triglyceride levels, and a recombinant betatrophin inhibited the LPL activity [2,13]. It was recently reported that betatrophin plays a role in the proliferation of pancreatic β-cells treated with an insulin receptor antagonist (S961). Yi et al. found that the number of β-cells overexpressing betatrophin in a mouse liver was increased 17-fold, and their quality increased 3-fold [3]. These investigators also demonstrated that betatrophin, once released into the blood, increases the quantity and quality of β-cells by binding to unknown receptors, and that the overexpression of betatrophin can enhance the glucose tolerance and reduce the fasting blood glucose (FBG) in mice. These findings are controversial. For example, mice with betatrophin deficiency exhibit normal blood glucose and glucose tolerance [14]. A study revealed that the proliferation of pancreatic β-cells in mice was unaffected by betatrophin. Further experiments confirmed that betatrophin deficiency reduced the blood TG level, opposite to its overexpression [5]. In short, the effects of betatrophin on the pancreatic β-cell proliferation in humans and mice are complex, but it is widely accepted by most scholars that betatrophin can enhance insulin resistance and the corresponding influence on lipid metabolism.

### 1.2. Structure of Betatrophin

The betatrophin gene, located on the 19p13.2 chromosomal region in humans and on chromosome 9 in mice, is primarily expressed in the liver and adipose tissues of humans, and in the liver, adipose, adrenal gland, and intestinal tract tissues in mice. The gene, encoding a secretory protein (approx 22,000-Dalton) consisting of 198 amino acids, contains 4 exons and is located in the intron region of the antisense strand of another gene, the dedicator of cytokinesis 6 (Dock6). Betatrophin is a highly evolutionarily conserved gene, and human betatrophin has a sequence consistency of 73% and structural similarity of 82% with mouse betatrophin. Predominantly expressed in the liver and adipose tissues, betatrophin is also found in human plasma, where it regulates the level of serum protein [3,15,16] and lipid metabolism. Moreover, the gene sequence of betatrophin has variously been named ‘refeeding induced in fat and liver’ (RIFL), lipasin, and ANGPTL [2,13,17].

According to research, the specific expression pattern of betatrophin in mammals manifests its physiological functions via temperature regulation as well as during pregnancy and lactation [18,19]. In mammals, the betatrophin protein sequence alignment illustrates that there are several partially conserved regions, and sequence analysis further uncovers that the N-terminal region of betatrophin contains the signal sequence of secreted or membrane-bound proteins [20]. Additionally, some protein modification sites are predicted to be in the N-terminal region of betatrophin, and the existence of a predicted casein kinase phosphorylation site further indicates that functional regulation of betatrophin activity is related to rapid signaling responses. These hypothetical modification sites are highly conserved among species from wallabies to humans. By and large, betatrophin sequences are highly conserved at the C-terminal end and bear conserved N-terminal regions in gorillas, sheep, and others. Thus far, the protection of betatrophin protein sequences has exerted vital effects, but the significance of specific betatrophin motifs in mammals remains to be determined.

### 1.3. Localization and Secretion of Betatrophin

It has been demonstrated that betatrophin is predominantly present in the cytoplasm and distributed as vesicles of various sizes [21]. The small dot-like betatrophin vesicles (≤1 μm) are usually solid and dispersed in the cytoplasm, while larger betatrophin vesicles (1–2 μm) become empty and are often related to lysosome-associated membrane protein 2 (LAMP2) and/or lipid droplet protein 2 (PLIN2), indicating the involvement of betatrophin in hydrolytic degradation or lipid regulation pathways. Researchers have found that the N-terminal sequence (1–20 amino acids) of betatrophin contains a predicted signal peptide, indicating protein secretion and its binding to the membrane [21]. Furthermore, betatrophin secretion in humans and mice was demonstrated to be correlated with serum TG or very-low-density lipoprotein (VLDL) concentration [4,22]. The association between intracellular betatrophin and lipid droplets implies that betatrophin can be used as a lipoprotein, which may be secreted or carry a lipid-related compartment [21].

## 2. Betatrophin and Metabolism

### 2.1. Betatrophin and Glucose Metabolism

Some scholars have endeavored to explore the function of betatrophin in glucose metabolism. In 2013, an in vivo study confirmed that mice with high expression of betatrophin had a lower level of FBG and superior glucose tolerance, revealing the potential relationship between betatrophin and glucose metabolism [3]. Nevertheless, there is still no consensus on whether betatrophin regulates glucose metabolism. Some mouse experiments have demonstrated that the postprandial blood glucose concentration and glucose tolerance are not changed by the high expression [5,23] or even the knockout [24] of betatrophin. Conversely, recent in vitro experiments showed that betatrophin overexpression in HepG2 cells promotes the synthesis of glycogens and strengthens the inhibition of insulin on the expression of PEPCK and G6Pase, two key enzymes involved in the process of gluconeogenesis. Hence, it is thought that betatrophin may facilitate glycogen synthesis and suppress gluconeogenesis in glucose metabolism [25]. The relationship between betatrophin and glucose metabolism was also supported by their in vivo experiment results. In addition, Wang et al. investigated the effect of the overexpression of betatrophin on glucose metabolism by transfecting HepG2 cells with betatrophin mRNA; they showed that the overexpression of betatrophin elevated the expression of glucose transporter 2 (GLUT2) [26]. Published in vivo and in vitro studies are not consistent in clarifying the role of betatrophin in reducing blood glucose and modulating glucose tolerance. As such, the physiological functions of betatrophin in the control of glucose metabolism need to be further interrogated.

### 2.2. Betatrophin and Lipid Metabolism

Work with animal models has shown that betatrophin affects adipose formation and that knocking out betatrophin reduces the level of TG in adipose tissue to levels lower than in wild mice, suggesting that betatrophin may be involved in the regulation of TG [13]. Gusarova et al. also confirmed that TG levels decrease after the expression of betatrophin is suppressed in mice. Betatrophin, a member of the ANGPTL family, interacts with ANGPTL3, therefore it is involved in the metabolic process of TG. The potential mechanism of betatrophin in increasing TG levels involves facilitating the cleavage of ANGPTL3 and releasing the N-terminal domain of ANGPTL3, thereby suppressing LPL and facilitating adipose decomposition. Inhibiting betatrophin downregulates the TG level in cells [5]. Work using mouse betatrophin knockout models has confirmed that the regulation of the plasma TG concentration by betatrophin is related to the activity of LPL [14]; additional research [18] has shown that TG levels are elevated only when both betatrophin and ANGPTL3 are present, implying that ANGPTL3 is necessary for the regulation of TG by betatrophin. To sum up, inhibition of LPL by betatrophin regulates TG metabolism, which may lay a foundation to target betatrophin for adipose regulation. Elevated TG levels in the liver cells of mice with expressed betatrophin also verify that betatrophin inhibits LPL, further revealing the mechanism of betatrophin in lipid metabolism [2]. Furthermore, the vital effects of betatrophin on lipid storage and metabolism have also been demonstrated via human experiments [4]. It is worth noting that Gómezambrosi et al.report that betatrophin exhibits a negative correlation with TG and a positive correlation with HDL [27]. This phenomenon may be attributed to a causal relationship between the overall level of betatrophin and lipid balance or improvement of lipase activity. In short, it appears that betatrophin modulates lipid metabolism, but the specific underlying mechanisms need further exploration.

## 3. Betatrophin and Insulin Resistance

### 3.1. Betatrophin and Obesity

Obesity leading to insulin resistance is the main risk factor for T2D. Specifically, obesity facilitates pancreatic β cell secretion and the utilization of blood glucose by tissues. When secretion by pancreatic β cells fails to satisfy insulin demand in vivo, β cell secretion will decline, and blood glucose will increase. Ultimately, this process will induce T2D [10]. During insulin resistance, betatrophin is predominantly derived from liver and adipose tissues. Betatrophin produced in the liver mainly participates in glucose metabolism, while that released from adipose tissues is correlated with inflammation, free fatty acids, and reduced adiponectin production [28,29]. Therefore, given the correlations between obesity, insulin resistance, and lipid metabolism, obesity is considered to affect the expression of betatrophin. In addition, it has been confirmed by animal experiments that obesity elevates betatrophin levels [2,13]. A recent population-based study demonstrated positive correlations between betatrophin and the body mass index (BMI) and the waist-to-hip ratio [11]. In another study, obesity was shown to up-regulate serum betatrophin levels while an exercise intervention reduced them [16]. However, there is some inconsistency in such populatin-based studies. An experiment conducted by Guo et al. [30] reported that betatrophin levels were higher in overweight individuals than in a control group, but obese individuals showed lower levels both the overweight and the control groups. In addition, Fenzl et al. [4] demonstrated no significant difference in the betatrophin levels between a normal control group and a morbidly obese group. A literature report has reported that betatrophin levels are reduced in obese people and even in those with insulin resistance [27]. The apparently contradictory results of the aforementioned studies may stem from uncontrolled variables related to disease complications, such as polycystic ovary syndrome and MetS, that themselves can regulate betatrophin levels in obese people [31,32,33].

### 3.2. Betatrophin and T2D

T2D, also referred to as adult-onset diabetes, is a highly prevalent disease frequently arising in patients aged over 35 years old; T2D patients account for over 90% of all diabetic patients. Most T2D patients are obese and lack proper insulin secretion, while also showing relatively low insulin sensitivity. Currently, the pathogenesis of T2D is thought to involve the following. The secretion of pancreatic β cells fails to meet the demand for insulin required by hyperglycemia and insulin resistance. Enhancement of β cell function to promote insulin secretion damages β cells over time and finally induces T2D [34,35,36]. Hence, drugs to promote the proliferation of β cells, increase insulin secretion, and improve insulin resistance have been extensively investigated [37,38]. The discovery of betatrophin as a hormone may be regarded as a scientific breakthrough to strengthen insulin resistance by raising the proliferation of β cells [38]. Work with animal models has shown that betatrophin can facilitate the proliferation of β cells and insulin production during insulin resistance [3]. Investigators have also found that betatrophin levels are increased in ob/ob mice and in db/db diabetic model mice. Whether betatrophin can actually enhance the proliferation ability of β cells has been questioned in subsequent studies [5,24]. Numerous population-based experiments showed that the levels of betatrophin in the T2D population are about 3 times higher than in the non-diabetic population [11] and that the risk of T2D in a population with a high level of betatrophin was increased by almost 6 times when age, gender, race, blood lipids, and other factors were controlled, indicating that the level of betatrophin can be used as an independent predictor of T2D. It has also been demonstrated betatrophin levels are positively correlated with age, BMI, FBG, HbA1c, and the waist-to-hip ratio. In addition, insulin was not increased and blood glucose was unaffected in T2D patients with high-levels of betatrophin. In addition, it was uncovered by Espes et al. [39] that the plasma betatrophin levels in T2D patients are markedly higher than that in non-diabetic individuals with the same gender, age, and other conditions. Fuand colleagues [23] also showed that plasma betatrophin levels rose in T2D patients and were correlated with blood glucose concentration. Erhan Onalan et al. [40] verified that circulating betatrophin and TNF-α levels were increased in MetS, IFG and IGT. The latest research by Seyhanli et al. [41] showed that betatrophin levels in the circulatory system of patients with gestational diabetes were higher than that of healthy people. In short, an increase of plasma betatrophin levels in T2D patients has been demonstrated. Conversely, others have shown that plasma betatrophin levels are lower in obese invidividuals, and in obese individuals with T2D [27]. Plasma betatrophin levels inversely correlate with insulin secretion capacity, suggesting that betatrophin levels are negatively regulated by insulin secretion capacity in humans [42]. Other experiments showed that overexpression of ANGPTL8 in db/db mice and mice with diabetes induced by high-fat diet/streptozotocin, reduces FBG levels and enhances glucose tolerance and insulin sensitivity, while ANGPTL8 knockdown has the opposite effect [43]. It is considered that these paradoxical results may stem from the use of different ELISA kits. Li et al. [44] comprehensively analyzed the relationship between T2D and plasma betatrophin levels by data retrieval from published work listed in PubMed and Embase, comparing the T2D and non-T2D populations. Nine studies conformed to the inclusion criteria and were included in the subsequent comprehensive analysis. The results of all the nine studies demonstrated that the levels of betatrophin rose in T2D population, and the levels of betatrophin in non-obese T2D individuals were higher than that in the non-T2D obese population. Additionally, there was no statistically significant difference in the levels of betatrophin between the non-T2D obese and T2D obese populations. However, they did not show that betatrophin levels were elevated in the T2D population that had been tested recently. Therefore, it is concluded that plasma betatrophin levels are stably increased in T2D population.

### 3.3. Betatrophin and MetS

MetS is a combination of metabolic disorders, such as obesity, hypertriglyceridemia, hyperglycemia, hypertension, and insulin resistance. The aforementioned MetS likely influences betatrophin levels [33,34]. It has been shown that betatrophin levels show positive correlations with obesity, blood lipid, and blood glucose in patients with MetS [32]. Maurer et al. [45] found that levels of betatrophin in obese patients increased with weight loss before and after fasting, so it was posited that there was a negative correlation between betatrophin and weight. However, it remains to be seen whether diet-based or surgical weight loss similarly affect betatrophin levels. Wang et al. [46] confirmed that except for blood pressure, the positive rate of the corresponding evaluation indices of MetS was up-regulated, and the level of betatrophin showed gradual consumption, leading to a conclusion that betatrophin was negatively correlated with MetS but not with blood pressure. In contrast, Zhai et al. [47] examined hypertension in the Chinese Han and Kazakhstan populations, and found that serum betatrophin was increased, suggesting that hypertension may affect the expression of serum betatrophin. The contradictory experimental results of betatrophin levels in individuals with hypertension complicated with MetS and those with only hypertension need further investigation. Recently it was shown that there is a correlation between the betatrophin/ANGPTL3/LPL pathway and the severity of coronary artery disease [48].

## 4. Influencing Factors for Betatrophin Expression

### 4.1. Nutrition

Researchers screened genes related to calorie intake in human adipocytes, including betatrophin and Dock6. The expression of betatrophin and Dock6 rose in the obese population administered a low-calorie diet, after ordinary food was reinstated, and in the healthy population who was overfed [49]. It was also shown that the gene expression of betatrophin in the liver, white adipose, and brown adipose tissues was induced in mice treated with a high-fat diet but inhibited in fasting mice [2,18]. Moreover, another study demonstrated that after fasting for 8–12 h, refeeding increased the expression of the betatrophin gene [18]. The levels of serum betatrophin protein were elevated within 2 h of a defined meal in a non-diabetic population [23]. These findings demonstrate that the expression of betatrophin gene and protein are associated with nutritional intake, and may fluctuate with experimentally induced food intake behavior.

### 4.2. Insulin

Compensatory insulin secretion is increased during obesity-induced insulin resistance; this is beneficial for blood glucose absorption to reduce blood glucose levels. Conversely, when blood glucose increases, insulin secretion is promoted increasing glucose absorption, utilization, and storage. As mentioned above, mice with a high expression of betatrophin have lower FBG levels and better glucose tolerance [3]. Therefore, it was speculated that betatrophin, insulin, and blood glucose may regulate each other. This was confirmed by work done using isolated hepatocytes by the team of Guo et al. [25] In their experiment, betatrophin gene and protein expression were examined following insulin treatment, showing that insulin dramatically elevated the expression of betatrophin. Additionally, it was confirmed that insulin could exert this promoting effect only in the presence of glucose and at the optimal concentration and stimulation time. It can be concluded that glucose is necessary for insulin to elevate betatrophin expression. An in vitro experiment using varying concentrations of glucose and insulin showed that high insulin concentration could up-regulate the expression of betatrophin in the case of insulin resistance [50]; the possible mechanism may be attributed to the insulin signaling pathway, that is, insulin binds to insulin receptors on the membrane to activate the phosphatidylinositol 3-kinase/protein kinase B (PI3K/Akt) signaling pathway, as it has been found that the use of PI3K/Akt blocker LY294002 inhibits the stimulating effects of insulin. The signaling molecular mechanisms behind how insulin increases betatrophin expression require further exploration. In addition, in vitro experiments of insulin treatment in patients with T2Dhave shown that exogenous insulin raises the level of betatrophin in patients with T2D compared to those of the same gender, age, blood lipid levels, and BMI, further demonstrating that insulin directly up-regulates betatrophin expression. However, research findings related to the relationship between insulin and betatrophin are not completely consistent. One study revealed that insulin was positively correlated with betatrophin in a non-diabetic group, but not related to betatrophin in a T2D group, although the T2D group exhibited an increased level of betatrophin [11,51]. In short, there is an ongoing debate about whether insulin affects the expression level of betatrophin directly or indirectly through blood glucose. Although this has been examined, there is still no reasonable explanation for the contradictory conclusions, which still require verification. In particular, more efforts should be made to explore these signaling mechanism.

### 4.3. Adiponectin

Adiponectin, the main adipocyte factor, exerts clinical and metabolic effects on adipose tissues, showing negative correlations with obesity and insulin resistance [52]. It was recently shown that adiponectin contributes to the improvement of metabolic efficacy [53]. According to one report, adiponectin, as the most abundant peptide secreted by adipocytes, is an effective regulator of lipid and glucose metabolism, with anti-diabetes, anti-atherosclerosis, and anti-inflammatory effects and with a vital role in the pathogenesis of metabolic diseases [54]. Recently, a negative correlation between the level of circulating betatrophin and adiponectin concentration was revealed. However, further work is required to elucidate adiponectin mechanisms [55]. The relationship between Adiponectin and Betatrophin, and their response to exercise and IR are showed in Figure 1. 

### 4.4. Exercise

It has been shown that exercise training is a kind of modulator and a potential therapeutic agent for obesity-triggered insulin resistance and diabetes [56,57]. Exercise enhances insulin sensitivity in obese individuals, and increases glucose intake by promoting the phosphorylation of Akt substrate AS160 in skeletal muscle [58]. It has also been found that betatrophin levels decline following exercise, but interestingly, this decline only occurs in the obese population [16]. This study supported the inhibitory role of betatrophin, and betatrophin was identified as a potential therapeutic target. In another experiment, a six-month calorie-limited diet or diet + exercise was used as a weight-loss measure, resulting in a dramatic reduction of betatrophin. This effect was achieved by body fat reduction, but not by physical activity [59]. The research of Purwo [60] illustrated that the level of betatrophin declined at 10 min after moderate interval training and moderate continuous training, and this decline was more evident at 10 min after moderate continuous training. In addition, the level of betatrophin before exercise also appeared to be positively correlated with obesity-related markers.

## 5. Conclusions

To sum up, betatrophin is crucial for the development of T2D, insulin resistance, and lipid metabolism, and its stimulation is usually associated with energy consumption or heat generation. In addition, the biological effects of betatrophin may be modulated by insulin, nutrition, adiponectin, and exercise, although the underlying molecular mechanisms, especially the signaling pathways of these intervention factors, require further experimental investigation.

## Figures and Tables

**Figure 1 metabolites-12-00925-f001:**
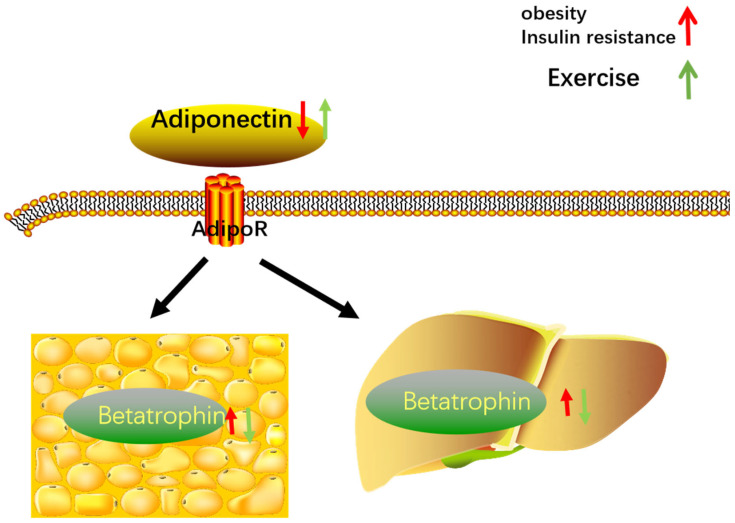
The relationship between Adiponectin and Betatrophin, and their response to exercise and IR.

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
