# Peer review of "Betatrophin and Insulin Resistance"

_metabolites, 2022, doi:10.3390/metabo12100925_

Round 1

Reviewer 1 Report

This should not be accepted for publication in its current form. It is extremely poorly written. Abstract and intro are identical. Several sentences are void of specific info. For example line 48 and 49: "In 2012, betatrophin was shown to be related to serum TG levels and to regulate lipase activity in mice [2,12]." ... this is a very frustrating sentence because the reader does not learn how it is related to TG levels and how LPL activity is regulated (does it increase or decrease LPL activity?). 

Author Response

Response to Reviewer 1 Comments

Point 1: Abstract and intro are identical.

Response 1: Thank you so much for reviewer’s reminding. The parts have been revised accordingly.

Point 2:  Several sentences are void of specific info. For example line 48 and 49: "In 2012, betatrophin was shown to be related to serum TG levels and to regulate lipase activity in mice [2,12]." ... this is a very frustrating sentence because the reader does not learn how it is related to TG levels and how LPL activity is regulated (does it increase or decrease LPL activity?).

Response 2: Thank you for your question. The statement has been revised accordingly for clarity.

Reviewer 2 Report

The review of Betatrophin is undeniably interesting, but it is advisable to include at least one figure in the manuscript to make it easier to understand and more attractive. I consider that the manuscript summarizes clearly the role of Betatrophin in IR. There is only a spelling mistake in the title, Betatrophin is misspelled. I also believe that the manuscript would benefit from having figures, at least one summarizing the interactions of Betatrophin with other parameters in glucose and lipid metabolism, obesity and diabetes. Lastly, there are studies on Betatrophin in bariatric surgery that would be of interest to comment in the obesity section. There is a mistake in the title that you should correct, Betatrophin is misspelled. 

Author Response

Response to Reviewer 2 Comments

Point 1: The review of Betatrophin is undeniably interesting, but it is advisable to include at least one figure in the manuscript to make it easier to understand and more attractive. I consider that the manuscript summarizes clearly the role of Betatrophin in IR.

Response 1: We really agree with the reviewer’s suggestion that the fig has been added accordingly.

Point 2:  There is only a spelling mistake in the title, Betatrophin is misspelled. I also believe that the manuscript would benefit from having figures, at least one summarizing the interactions of Betatrophin with other parameters in glucose and lipid metabolism, obesity and diabetes.

Response 2: Thank you so much for reviewer’s reminding. The parts have been revised accordingly.

Point 3:Lastly, there are studies on Betatrophin in bariatric surgery that would be of interest to comment in the obesity section. There is a mistake in the title that you should correct, Betatrophin is misspelled.

Response 3: Thank you for your suggestion.We have revised this issue to the best of our ability.

Reviewer 3 Report

Abstract and Introduction cannot be the same sentence. Should be changed to appropriate text.

Although it is a review article, it should be easier for the reader to understand if there is at least one coherent figure.

In this article, it will find some parts with no spaces between words. Authors should be handled properly.

Author Response

Response to Reviewer 3 Comments

Point 1: Abstract and Introduction cannot be the same sentence. Should be changed to appropriate text.

Response 1: We really agree with the reviewer’s suggestion that the Instroduction has been revised accordingly.

Point 2:Although it is a review article, it should be easier for the reader to understand if there is at least one coherent figure.

Response 2: We really agree with the reviewer’s suggestion that the fig has been added accordingly.

Point 3:In this article, it will find some parts with no spaces between words. Authors should be handled properly.

Response 3: Thank you so much for reviewer’s reminding. The parts have been revised accordingly.

Reviewer 4 Report

The review from Guo, et al., discusses about betatrophin which is involved in development of type-2-diabetes, obesity, etc due to its effects on insulin, adiponectin, etc. The work has reviewed existing literature on the understanding of betatrophin structure, localization, effect on glucose and lipid metabolism, etc to show the lacunae that exist in the field regarding betatrophin. There have been previous reviews published elsewhere such as J-Z Zhu, et al., BioMed Rep, 2014; M. Abu-Farha, et al., Diabetes Metabolism Research and Reviews, 2017, etc which might be less detailed on this topic though.

1)      Previous reviews in the area as J-Z Zhu, et al., BioMed Rep, 2014; M. Abu-Farha, et al., Diabetes Metabolism Research and Reviews, 2017 needs to be cited appropriately.

2)      Citation in the field all through the review has not been consistent many of leading studies in the field Takebayashi, et al., J Clin Med Res; S. Tokumoto, et al., Diabetic Medicine, 2015 while discussing about serum betatrophin; Siddiqa, et al., Computational Biology and Chemistry, 2016; etc and many have been ignored.

3)      Although a lot of concepts have been discussed all along the review there are no figures to put this into perspective. Figures help in better understanding of the science that the authors want to convey. Especially parts such as insulin signalling, insulin secretion, adiponectin signalling described in the review can be better represented with cartoons to show the intracellular processes.

4)      At many places there are words without spaces some examples are: 191 line - numerouspopulation in page 4, 226 line aforementionedMetSlikely in page 5, 235 line in withMetS in page 5, 240n line in Reently[44]it in page 5. These are just some examples from a section of the write up and lot of such examples exist in each page and must be reviewed.

Author Response

Response to Reviewer 4 Comments

The review from Guo, et al., discusses about betatrophin which is involved in development of type-2-diabetes, obesity, etc due to its effects on insulin, adiponectin, etc. The work has reviewed existing literature on the understanding of betatrophin structure, localization, effect on glucose and lipid metabolism, etc to show the lacunae that exist in the field regarding betatrophin. There have been previous reviews published elsewhere such as J-Z Zhu, et al., BioMed Rep, 2014; M. Abu-Farha, et al., Diabetes Metabolism Research and Reviews, 2017, etc which might be less detailed on this topic though.

Point 1: Previous reviews in the area as J-Z Zhu, et al., BioMed Rep, 2014; M. Abu-Farha, et al., Diabetes Metabolism Research and Reviews, 2017 needs to be cited appropriately.

Response 1: We really agree with the reviewer’s suggestion that the references has been revised accordingly.In the 7th and 20th references, respectively

Point 2:  Citation in the field all through the review has not been consistent many of leading studies in the field Takebayashi, et al., J Clin Med Res; S. Tokumoto, et al., Diabetic Medicine, 2015 while discussing about serum betatrophin; Siddiqa, et al., Computational Biology and Chemistry, 2016; etc and many have been ignored.

Response 2: Thank you for your reminding.We added some of these references.

Point 3:  Although a lot of concepts have been discussed all along the review there are no figures to put this into perspective. Figures help in better understanding of the science that the authors want to convey. Especially parts such as insulin signalling, insulin secretion, adiponectin signalling described in the review can be better represented with cartoons to show the intracellular processes.

Response 3: We really agree with the reviewer’s suggestion that the fig has been added accordingly.

Point 4: At many places there are words without spaces some examples are: 191 line - numerouspopulation in page 4, 226 line aforementionedMetSlikely in page 5, 235 line in withMetS in page 5, 240n line in Reently[44]it in page 5. These are just some examples from a section of the write up and lot of such examples exist in each page and must be reviewed.

Response 4: Thank you for your reminding.We have revised this issue to the best of our ability.

Round 2

Reviewer 3 Report

All concerns are improved.

Reviewer 4 Report

Thank you very much for addressing the listed comments. A spell check and minor editing might be required to sort out spaces. Otherwise, this looks fine.